# Ice rule fragility via topological charge transfer in artificial colloidal ice

András Libál[1,2], Dong Yun Lee[3], Antonio Ortiz-Ambriz [3,5], Charles Reichhardt[1], Cynthia J.O. Reichhardt[1], Pietro Tierno [3,4,5] & Cristiano Nisoli[1,6]

Artificial particle ices are model systems of constrained, interacting particles. They have been introduced theoretically to study ice-manifolds emergent from frustration, along with domain wall and grain boundary dynamics, doping, pinning-depinning, controlled transport of topological defects, avalanches, and memory effects. Recently such particle-based ices have been experimentally realized with vortices in nano-patterned superconductors or gravitationally trapped colloids. Here we demonstrate that, although these ices are generally considered equivalent to magnetic spin ices, they can access a novel spectrum of phenomenologies that are inaccessible to the latter. With experiments, theory and simulations we demonstrate that in mixed coordination geometries, entropy-driven negative monopoles spontaneously appear at a density determined by the vertex-mixture ratio. Unlike its spin-based analogue, the colloidal system displays a "fragile ice" manifold, where local energetics oppose the ice rule, which is instead enforced through conservation of the global topological charge. The fragile colloidal ice, stabilized by topology, can be spontaneously broken by topological charge transfer.

[1] Theoretical Division, Los Alamos National Laboratory, Los Alamos, NM 87545, USA. [2] Mathematics and Computer Science Department, Babeș-Bolyai University, Cluj 400084, Romania. [3] Departament de Física de la Matèria Condensada, Universitat de Barcelona, Barcelona 08028, España. [4] Universitat de Barcelona Institute of Complex Systems (UBICS), Universitat de Barcelona, Barcelona 08028, Spain. [5] Institut de Nanociència i Nanotecnologia, Universitat de Barcelona, Barcelona 08028, Spain. [6] Institute for Materials Science, Los Alamos National Laboratory, Los Alamos, NM 87545, USA. Correspondence and requests for materials should be addressed to C.N. (email: cristiano@lanl.gov)

The ice rule[1] has a long, fascinating history that has influenced thermodynamics, physical chemistry, statistical mechanics, magnetism, materials science, and soft matter. In the 1930s Giaque and Ashley[2,3] found that the specific entropy of water at very low temperature was not zero, despite the ordered, solid structure of ice. In water ice the oxygen atoms reside at the center of tetrahedra, sharing four hydrogen atoms with four nearest neighboring oxygen atoms. Two hydrogen atoms are covalently bound to each oxygen, and two form hydrogen bonds with neighboring oxygen atoms. In the so-called ice rule introduced by Bernal and Fowler[1], this situation is described as having two hydrogen atoms pointing "in", and two pointing "out" of the tetrahedron. As Linus Pauling explained[4], the freedom in choosing such an arrangement on a large lattice leads to a degeneracy that grows exponentially with the number of tetrahedra, generating the residual entropy.

This idea proved to be not limited to water. The ice rule was recognized to occur in exotic magnets, namely the rare earth titanates such as $Ho_2Ti_2O_7$ and $Dy_2Ti_2O_7$[5,6]. These pyrochlore systems were called "spin ices" because the cations $Ho^{3+}$ and $Dy^{3+}$ carry a large magnetic moment directed along the lattice bonds, which can be associated to a classical, binary Ising spin. At low temperature, frustration ensures that two spins point in and two out of each vertex, reproducing the ice rule and preventing the spontaneous magnetization of the material.

The ice rule was eventually exploited to design new artificial frustrated systems based on magnetic nano-islands, confined colloidal particles, or vortices in superconductors[7–26] that generalize spin ices and are broadly called artificial spin ices. There, exotic states of matter and emergent dynamics often not found in natural systems can be deliberately designed and externally controlled in artificial nano- and micro-scale materials.

In such systems frustration produces complex disordered manifolds where fascinating effects, such as dimensionality reduction[27], emergent descriptions[28–30], topological constraints[31,32], and complex dynamics of magnetic (or more generally topological) charges[17,33,34] can be tailored, nano- or micro-engineered, and characterized at the level of the constitutive degrees of freedom, often providing remarkable vistas of statistical mechanics in action[35–37]. Such generality is not surprising since the ice-rule is a powerful topological prescription for conceptualizing the effects of frustration in a broad class of physical systems.

To understand the topological nature of the ice rule in the broadest generality, consider a general lattice, even a graph, or network[38] with nodes of various coordination number $z$. Assign binary variables on the edges of the graph, in the form of spins directed along the edges and impinging in the nodes. Then we can define a "topological charge" $q$ for each vertex as the difference in the number of spins $n$ pointing toward the vertex and the number of spins $z - n$ pointing away from it, or $q = 2n - z$. In magnetic spin ices $q$ is proportional to the magnetic charge of the vertex[39,40] leading to a rich phenomenology for magnetic charge currents[41], charge ordering[34,42,43], charge screening[28,33] or dynamical arrest[44]. In this language, the ice rule corresponds to the minimization of the absolute value of the topological charge $|q|$. The charge is called "topological" insofar as it depends upon the connectivity of the system, and its definition does not change for continuous deformations of the lattice. It is therefore a topological invariant for the vertex configuration (though it does not completely define the spin configuration[30,34,45]). On vertices of even coordination the minimization of $|q|$ on each vertex implies that $q = 0$, and when $z = 4$ we recover the original 2-in/2-out ice rule of water ice. On vertices of coordination $z = 3$, the minimum occurs for $q = \pm 1$, corresponding to 2-in/1-out or 1-in/2-out.

In the magnetic spin ice-like systems mentioned above, the low energy ensembles all obey the ice rule, which has proven to be extremely robust. The ice rule survives all sorts of weak or strong alterations, including decimation[29], mixed coordination[27,28], and the introduction of dislocations;[32] indeed it was found that even isolated clusters of magnetic vertices obey the ice rule at low energy[46].

Here we add to the already rich history of the ice rule by introducing a system where the ice rule becomes "fragile", meaning that it can be easily destabilized by topology. Through a combination of theory, simulations, and experiments, we demonstrate that the colloidal ice falls in a different class of geometrically frustrated ices, or "fragile ices." There, the ice rule is spontaneously broken in lattices of mixed coordination, leading to a rich and unique set of phenomena, including topological charge transfer and charge screening, that are completely absent in nanoscale magnetic ices or indeed in most ice systems known to us. It is important to understand that we are describing fragility, not a breakdown. As we will see, most of the system still obeys the ice rule, and only specific charges, in the form of negative monopoles, appear. Crucially, these monopoles are not excitations, but instead belong to the low energy state, and their density can be controlled.

## Results

**The system.** The content of this article can be summarized by referring to its figures. In Figs. 1, 2 we illustrate the system: repulsive colloids are gravitationally trapped in microgrooves with two preferential positions at the extremes, making each groove equivalent to a binary Ising spin. The grooves are arranged along the edges of a square lattice, the colloids repel each other, and the system obeys the ice rule (Fig. 3a), as already found in Refs.[11–13,23]. When we decimate our system by removing colloids (Fig. 2), we obtain a lattice of mixed $z = 3,4$ coordination. There, we observe, the ice-rule is spontaneously yet selectively violated as negative $q = -2$ charges form on the $z = 4$ vertices (Figs. 3, 4). The $z = 3$ vertices still all obey the ice rule; however, the relative ratio of $q = 1$ to $q = -1$ charges changes in order to compensate the negative charge of the $z = 4$ vertices (Fig. 5), since the total topological charge of a system of "dipoles" must remain zero. This global fragility of the ice rule introduces further local phenomenology, as charges also rearrange locally to screen the $q = -2$ monopoles appearing on $z = 4$ vertices (Fig. 5). This happens because, as previously noted by one of us[47,48], the ice rule in magnetic spin ice systems is enforced locally by the vertex energetics, but globally in colloidal spin ices, by the conservation of topological charge. In fact, in colloidal systems the ice rule is actually opposed by the local vertex energy, as we explain below. Since magnetic ices are locally at an energy minimum, they are structurally "robust" ices. In contrast, the colloidal ice has a collective low-energy manifold that is composed of an energetic compromise between locally excited vertices and is thus a "fragile" ice. Since the resulting energetically unstable arrangement is stabilized by topology, it can also be easily and deliberately destabilized through topology to create new emergent states.

The system under study is shown schematically in Fig. 1. We start from an array of bistable traps arranged along the edges of a square lattice. Each trap contains a colloid, gravitationally confined, that can preferentially occupy the two ends of the traps. The colloids are paramagnetic and can be magnetized by a field perpendicular to the plane of the array, introducing repulsive, isotropic, colloid-colloid repulsion. This system is known to obey the ice rule[11,23,24].

We then consider a "decimation" of such an array, in which we remove certain traps (or, equivalently, certain colloids from the

traps) in a random fashion, in order to create a lattice of mixed coordination $z = 3, 4$. The result is a decimated square array of traps as shown in Fig. 2(a). Without any decimation protocol, the simple elimination of traps at random from the structure would create vertices of coordination $z = 3$, $z = 2$, and $z = 1$. To reduce complexity, however, we prefer to generate only $z = 3$ vertices through decimation (although our considerations also apply to other cases where $z = 2$ and $z = 1$ vertices are present[47,48]). We achieve our decimation using a partial, random dimer covering of the edges (Fig. 2(a)), where randomly chosen edges are covered by dimers in such a way that each vertex is covered by at most

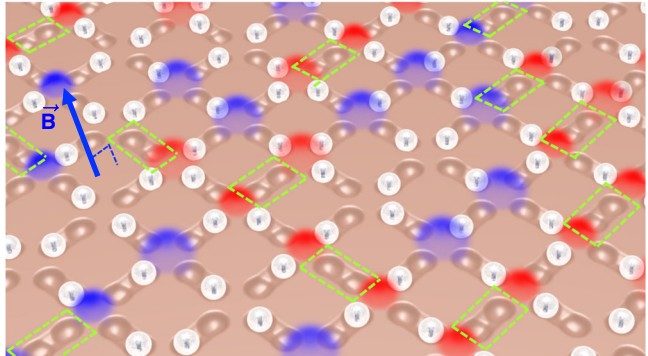

**Fig. 1** Schematic of the system. The experimental system consists of paramagnetic colloids placed via optical tweezers in lithographic double wells arranged along the edges of a square lattice. Each colloid is gravitationally trapped in one microgroove, and it can sit in one of the two wells. A perpendicular field **B** magnetizes the colloids, thus introducing a repulsive dipolar interaction. The edges of the square lattice can be decimated by simply removing the colloids from the corresponding microgrooves (dashed green rectangles). Red and blue glows denote positive and negative topological charges, respectively

one dimer. We then remove an edge between two "dimerized" vertices of coordination $z = 4$, in order to obtain only vertices of coordination $z = 3$.

We introduce some nomenclature that will prove to be useful later. Considering the thermodynamical limit of the system, and neglecting boundary effects, we call $N_t$ the number of traps in the original square lattice, which form a total of $N_v = N_t/2$ vertices of coordination $z = 4$ (Fig. 2(a)). We decimate the lattice by removing $N_d$ traps, in accordance with the dimer model protocol. Each time a trap is removed, two $z - 4$ vertices change into two $z = 3$ vertices. We call $N_{z_3}$ and $N_{z_4}$ the resulting number of vertices of coordination $z = 3$ and 4, respectively. The decimation density is defined as $\xi = N_d/N_t$, while $\eta = N_{z_3}/N_{z_4}$ is the ratio between the two vertex types. Our dimer-cover based decimation strategy thus gives $N_{z_4} = N_v - 2N_d$, $N_{z_3} = 2N_d$, and therefore $\eta = 4\xi/(1 - 4\xi)$.

The maximum possible decimation corresponds to a complete random dimer covering realized when all the vertices are covered by one and only one dimer. Then the number of dimers is half of the number of vertices and therefore a quarter of the number of traps. Thus the maximum decimation corresponds to a removal of 25% of the traps, or $\xi = 1/4$. Note that $\eta \to +\infty$ when $\xi \to 1/4^-$, since $N_{z_4} = 0$ at this maximal decimation: all vertices have coordination $z = 3$.

Figure 2(b) shows the energetics of the resulting vertices of coordination $z = 3$ and $z = 4$ arranged in order of increasing energy, which also corresponds to increasing topological charge. Note that in computing the vertex energy we adopt a nearest neighbor approximation and consider only the interaction of the particles close to the vertex. Vertices of coordination $z = 4$ can have even charges $q = -4, -2, 0, 2, 4$, whereas vertices of coordination $z = 3$ can have odd charges $q = -3, -1, 1, 3$. We label the vertices by their topological charge, and call $N_{z_4,q}$ and $N_{z_3,q}$ the number of vertices of charge $q$ and coordination $z = 4,3$, respectively. We define the relative vertex frequencies as $n_{z_4,q} = N_{z_4,q}/N_{z_4}$ and $n_{z_3,q} = N_{z_3,q}/N_{z_3}$.

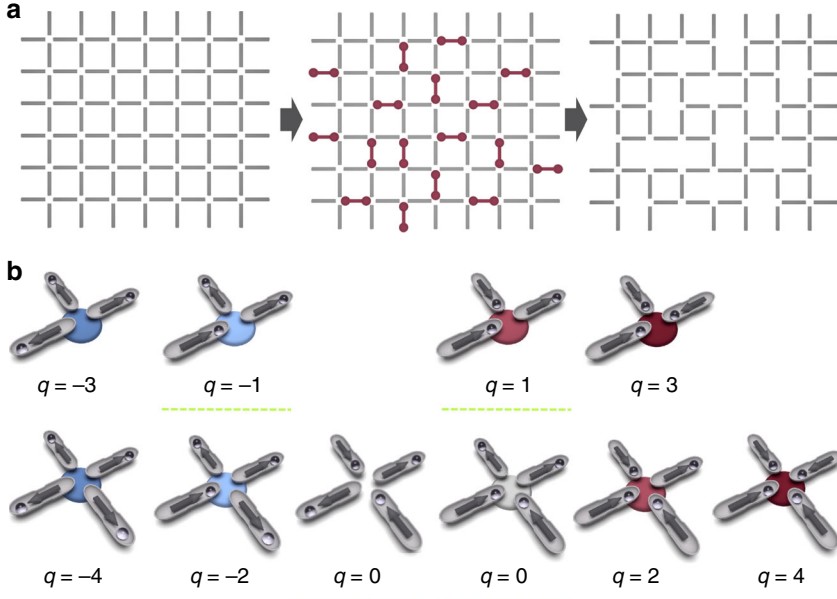

**Fig. 2** Schematics of the decimation. **a** A decimation of the square lattice that creates only $z = 3$ and $z = 4$ vertices, but no $z = 2$ vertices, is equivalent to a partial dimer covering (red dumbbells) of the edges. **b** Colloid configurations for vertices of coordination $z = 4,3$, in order of increasing topological charge and thus energy. The ice rule vertices have minimal absolute charge, which is $q = \pm 1$ for $z = 3$ vertices and $q = 0$ for $z = 4$ vertices (dashed green underline), and yet, unlike in magnetic spin ice, their energy is not the lowest. Red (blue) disks denote positive (negative) charges. A gray disk on a $z = 4$ vertex indicates a zero charge excitation corresponding to a biased ice rule vertex. The vertex without a disk represents the "ground state" of the square ice. Arrows aligned along the groove and pointing toward the colloids represent the analogy with a spin ice system

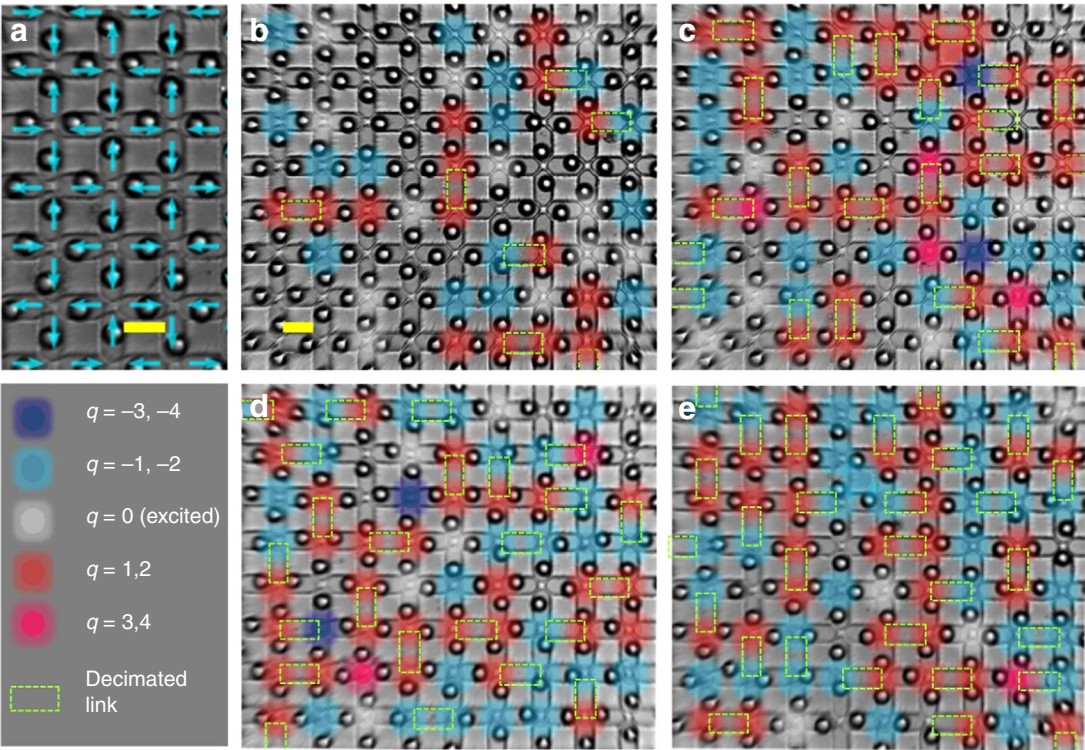

**Fig. 3** Experimental results. **a** An experimental image of the undecimated system shows the expected antiferromagnetic ordered configuration. The blue arrows denote spins associated with the double wells occupied by the colloids. **b–e** Experimental images of the colloidal system at increasing decimation corresponding to (**b**) $\eta = N_{z_3}/N_{z_4} = 0.19$, $\xi = N_d/N_v = 0.04$, (**c**) $\eta = 1.3158$, $\xi = 0.142$, (**d**) $\eta = 2.3846$, $\xi = 0.176$ and (**e**) $\eta = 5.2857$, $\xi = 0.21$. Dashed green rectangles denote decimated traps corresponding to $z = 3$ vertices. Negative charges of $q = -2$ monopoles (blue glows) form on $z = 4$ vertices as a consequence of decimation, in violation of the ice rule. Meanwhile $z = 3$ vertices still obey the ice rule, but develop an overabundance of $q = -1$ charges to adsorb the negative charge of the $z = 4$ vertices. This phenomenon increases as decimation increases. Scale bars (yellow) are 20 μm. See Supplementary Movies 1 and 2 for corresponding movie clips

We have performed experiments on mixed-coordination lattices at various levels of decimation. We corroborate our experimental results using overdamped Langevin dynamics on larger samples. For both experiments and simulations, at each level of decimation the results are obtained by averaging over ten different, randomly generated lattices obtained via the random dimer algorithm described above.

**Experimental and numerical results**. The experimental system is based on a monolayer of paramagnetic colloids confined above a square lattice of lithographic, microscopic double-wells (Fig. 1). Each gravitational trap permanently confines a colloid, and contains a small central hill that the colloid can cross under the influence of colloid-colloid interactions (see Methods). We apply an external magnetic field **B** perpendicular to the plane to induce a tunable, perpendicular dipole moment $\mathbf{m} \propto \mathbf{B}$ in each colloid. The resulting interaction between two colloids a distance $r$ apart is repulsive, isotropic, and given by $U_d \sim m^2/r^3$. We use optical tweezers to load one colloid into each double-well, or to eliminate colloids from the traps during lattice decimation. Using video microscopy and particle tracking, we extract real-time dynamics and visualize the collective low-energy configurations. As the field **B** increases, so does the mutual repulsion, and the colloids, originally disposed randomly, rearrange to a collective low-energy configuration[23,24]. Our experimental system extends over a square lattice composed by $11 \times 8$ vertices corresponding to a total of $N_t = 195$ traps in the undecimated case. We note that the size of the experiments is limited by two factors: the trapping

objective constrains the field of view, and the time required to populate the system must be small enough to keep the suspension electrostatically stable.

In Fig. 3 we show snapshots of experimental results for different decimations. At zero decimation, the system obeys the ice rule, as shown in Fig. 3(a). At nonzero decimation, the ice rule is broken in the $z = 4$ sublattice, but very specifically: only $q = -2$ charges appear spontaneously, while all other vertex type follow the ice rule. At the same time, the ice rule is still obeyed on the $z = 3$ sublattice, where only charges $q = \pm 1$ are present.

We corroborate these experimental findings with numerical simulations on larger samples (2500 vertices with periodic boundary conditions) than those used in experiments (which contained only 88 vertices). We employ over-damped, Brownian dynamics precisely parametrized to mimic the experimental setting (see Methods and ref[17,49].). In Fig. 4 we show snapshots of the results of simulations at different decimation levels. Exactly as in the experiment, the simulations confirm that breakdown of the ice rule occurs on $z = 4$ vertices only. Furthermore, on a larger scale we find that the disordered charge transfer between $z = 4$ and $z = 3$ vertices implies the breakdown of the well known antiferromagnetic order[8,11] of the square ice manifold. This structural transition to disorder has been recently proved theoretically[48], but here we experimentally observe it at a decimation of approximately 12% of the traps, about halfway to the maximal decimation of 25%.

In Fig. 5 we provide a quantitative analysis of the numerical and experimental results along with our theoretical predictions, which are described later. In Fig. 5(a, b) we plot the relative

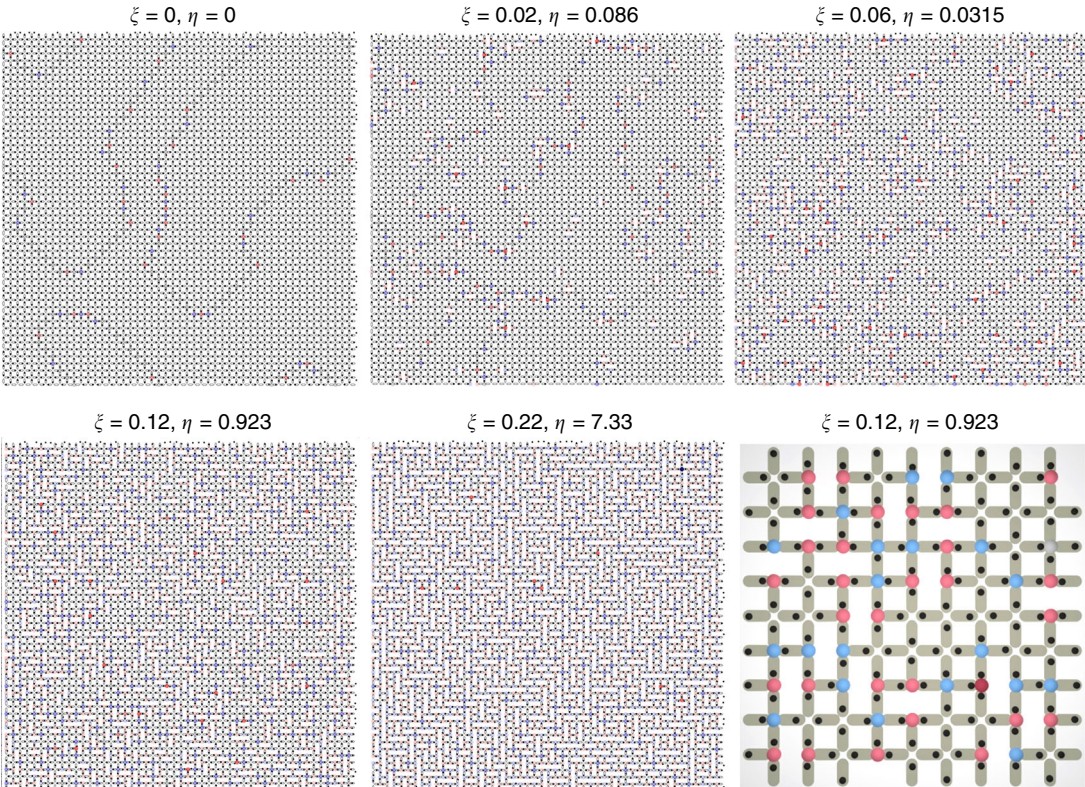

**Fig. 4** Numerical results. Snapshots of numerical simulations for increasing decimation with color coding as in Fig. 2(b) indicating vertex charges. At zero decimation ($\eta = 0$) large regions of the expected antiferromagnetic order separated by domain walls are visible. At low decimation of 2–6% ($\eta = 0.086, \eta = 0.315$), almost all of the $z = 3$ vertices are positively charged, while negative charges ($q = -2$) that appear on the $z = 4$ vertices can pin the domain walls, causing the ordered domains to shrink. At a decimation of 12%, there is already no discernible order, while at high decimation, about half of the $z = 4$ vertices violate the ice rule and host positive charges, which destroy the remaining ordering. In the zoomed portion of the $\xi = 12\%$ and $\eta = 0.923$ sample, the colloidal positions are visible and show details of violation of the ice rule at $z = 4$ vertices by negative, $q = -2$ monopoles only, but little or no ice rule violation at $z = 3$ vertices. See Supplementary Movies 3–6 for corresponding movie clips

frequencies $n_{z_4,q}$ and $n_{z_3,q}$ of vertices grouped by topological charge versus $\eta = N_{z_3}/N_{z_4}$, the ratio between the two vertex coordinations. Figure 5(a) shows more precisely that in the $z = 4$ sector vertices obey the ice rule, with the only violations arising from negative topological monopoles of charge $q = -2$. These negative charges appear spontaneously and increase in relative number as the amount of decimation increases, which increases the strength of the violation of the ice rule on vertices of coordination $z = 4$. A measure of ice rule violation, the total density of negative charge $q_{z_4} = \sum_q n_{z_4,q} q$ appearing on the $z = 4$ vertices, is plotted in Fig. 5(c) as a function of the lattice decimation.

Remarkably we find that the $z = 3$ vertices (Fig. 5(b)) always obey the ice rule, as was theoretically proposed in[47,48]. Indeed, only charge $q = \pm 1$ vertices are present for all but the very lowest decimations, with small deviations at $\eta < 1$ (see later). Figure 5(b) also shows that $q = 1$ vertices always exceed $q = -1$ vertices in number and thus the $z = 3$ vertices have an overall positive charge. They can therefore adsorb the extra negative charge introduced by the $z = 4$ vertices without leaving the ice-manifold simply by shifting their relative ratio in favor of vertices of charge $q = 1$. Moreover, Fig. 5(b) indicates that as $\eta = N_{z_3}/N_{z_4}$ tends to infinity (which means that the density of $z = 4$ vertices tends to zero), the fraction of vertices of charge $q = 1$ and $q = -1$ both tend to the same value of 1/2 as expected in a single coordination, $z = 3$ lattice.

Small deviations from this picture only happen at $\eta < 1$. There, $z = 3$ vertices are sparse and surrounded by $z = 4$ vertices. The

density of $z = 3$ is too small to adsorb all the available charge coming from the $z = 4$ and therefore the numerical simulations show larger $q = +3$ charges forming on them. In the experimental data only, we also see very few $q = +2$ monopoles forming on $z = 4$ vertices at low decimation. This is likely a consequence of lack of complete equilibration at low decimation, where the system is close to order, and of finite size of the sample, where positive charges can form on the boundaries as explained in the next subsection and in ref.[48]. Indeed, this type of defects was also present in previous work on non-decimated, ordered square lattice systems[23].

Moving away from the global picture, disorder of the ensemble produces fascinating local effects of spontaneous screening of topological charge. In Fig. 5(d) we plot $Q_{NN}$, the average charge neighboring a $z = 4$ vertex. We find that negative $q = -2$ monopoles are surrounded by a positive average charge that largely exceeds the average charge surrounding non-charged $z = 4$ vertices. Thus, as monopoles of charge $q = -2$ spontaneously appear on $z = 4$ vertices, they are screened by positive charges $q = 1$ on the surrounding vertices of coordination $z = 3$. This suggests that charge screening is not unique to magnetic charges that interact via a Coulomb law in magnetic ices[28,33,34,45]. In fact, charge rearrangement and ordering was also recently observed numerically in the disordered ensemble of kagome colloidal ice[49]. There, it was shown, it is a consequence of the $1/r^3$ long range interaction among colloids, the same present in this work. Charge effects driven by charge-charge interactions were also seen in

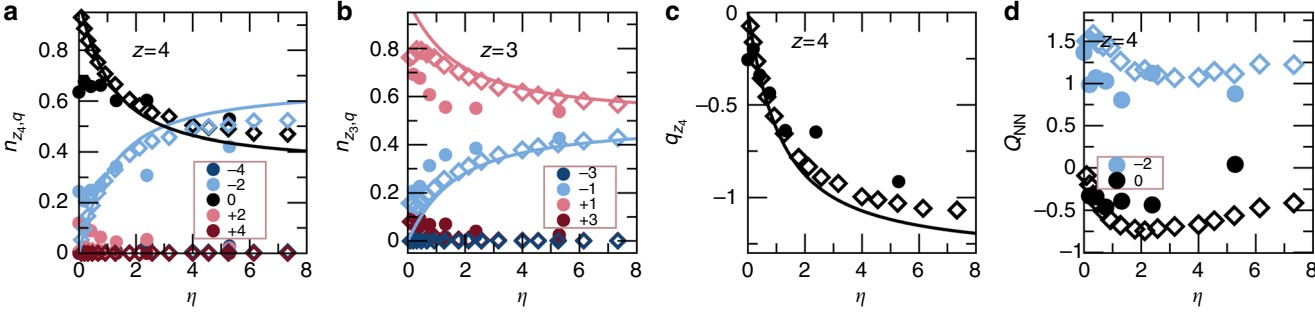

**Fig. 5** Comparison between experiments, numerics and theoretical predictions. Panels show experimental results (bullets) and numerical results (diamonds) compared to theoretical predictions (solid lines). **a** Vertex statistics $n_{z_4,q}$ at equilibrium vs $\eta = N_{z_3}/N_{z_4}$ for $z = 4$ vertices grouped by topological charge $q$. Dark blue: $q = -4$; light blue: $q = -2$; black: $q = 0$; pink: $q = +2$; red: $q = +4$. All the non-ice-rule vertices are suppressed except $q = -2$ monopoles, which increase with $\eta$ as the availability of $z = 3$ vertices for charge transfer increases. **b** Vertex statistics $n_{z_3,q}$ vs. $\eta$ for $z = 3$ vertices. Dark blue: $q = -3$; light blue: $q = -1$; pink: $q = +1$; red: $q = +3$. Only ice rule vertices are present ($q = \pm1$), but there is an excess density of positive $q = +1$ charges in order to screen the charge transfer from the $z = 4$ sector. As $\eta \to \infty$, the $z = 4$ sector disappears and thus $n_{z_3,q=1}$ and $n_{z_3,q=-1}$ tend to the same value of $n_{z_3,q=1} = n_{z_3,q=-1} = 1/2$, as also found in kagome ice[45]. **c** Net density of charge $q_{z_4}$ forming on $z = 4$ vertices vs $\eta$ as a measure of ice rule violation. **d** Charge screening $Q_{NN}$ of $q = -2$ monopoles (blue) and "screening" of $q = 0$ ice rule vertices (black) on $z = 4$ vertices vs $\eta$. At large decimation we find sparse $z = 4$ vertices embedded in a background of $z = 3$ vertices. The $z = 3$ vertices have an average charge of $\langle Q_3 \rangle = +0.15$, but the charge is much larger ($\langle Q_3 \rangle = +1.25$) in the nearest neighborhood of $q = -2$ monopoles. This indicates that the disordered sea of $z = 3$ charges screens the monopoles

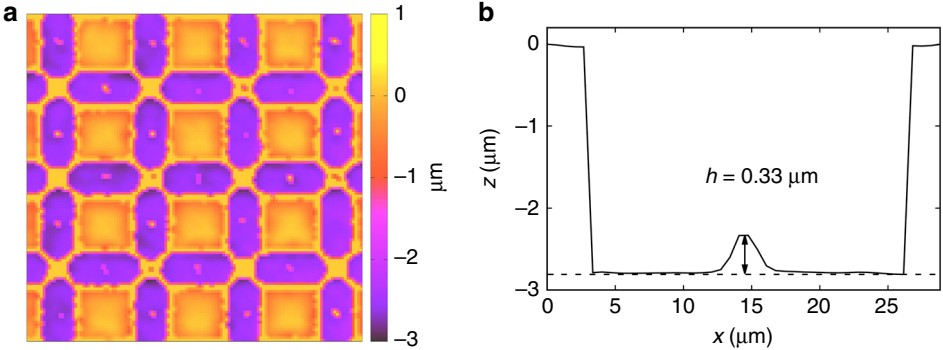

**Fig. 6** Experimental double well structure **a** Optical profilometer image of the square lattice of double wells after the lithographic process. **b** Profile of one double well characterized by a central hill of elevation $h = 0.33\,\mu m$

magnetic ice systems: charge ordering within the ice state of kagome artificial spin ice[34,42], and monopole screening by magnetic charges in Shakti ice[28]. Unlike in Shakti, here charges screen not excitations, but rather monopoles which belong to low energy state.

These results unambiguously demonstrate the breakdown of the ice rule in particle based ice as suggested in Ref.[47], along with non-trivial rearrangement of the topological charge being transferred. We reiterate that such a breakdown is not possible in magnetic spin ice systems, where the ice manifold has been shown to be completely robust[27–29,32,46].

**Entropic nature of ice rule fragility.** To understand the nature of the ice rule breakdown in colloidal ice, we first need to understand its origin, as the former differs essentially from the magnetic ices. In magnetic spin ice the topological charge minimization that corresponds to the ice rule is enforced by the local energetics, since the energy of frustrated spins meeting at a vertex typically scales quadratically with the vertex charge, or $E \sim q^2$ (ignoring geometrical effects[39]). In colloidal ices, however, the energy of $s$ repulsive colloids in a vertex scales as $E_s \sim s(s-1)/2 \sim q_s^2/8 + q_s(z-1)/4$[47], thus favoring vertices of large, negative charge, in violation of the ice rule. Obviously, the total charge must be zero, so it is not possible for all vertices to be

negatively charged. Individual vertices can only push their charge to the boundaries, and the resulting charge accumulation is limited by the size of the edges. Therefore, the density of topological charge in the bulk must scale at least as the reciprocal length of the boundaries, leading to the emergence of the ice rule (zero charge) in the thermodynamic limit. There is thus a collective, non-local reason for the ice rule in colloidal systems, which is quite unlike the local, energy-enforced origin of the ice rule in magnetic systems. Indeed, the latter is observed locally even in small spin ice clusters[46].

The boundary size constraint is lifted in our decimated system, since the $z = 4$ vertices now have an internal boundary consisting of $z = 3$ sub-lattices onto which topological charges can be pushed. Because the global charge must remain zero, the two sub-lattices develop opposite nonzero charges. As a consequence, the ice rule is very selectively broken in the $z = 4$ sector by the appearance of negative charges $q = -2$, corresponding to 1-in/3-out vertices. The ice rule still applies to the $z = 3$ vertices, since the plasma of charges in an odd-coordination spin ice can absorb and screen charges without breaking the ice rule.

We now make these considerations more quantitative[47]. For simplicity, we can treat vertices as uncorrelated, but constrain the total charge to be zero. Then the thermodynamic ensemble at equilibrium still follows a Boltzmann distribution but in the

effective vertex energies $\tilde{E}_s = E_s - q_s\phi$, where $\phi$ is a Lagrange multiplier enforcing the requirement of zero total charge. Thus, for a lattice of coordination $z$, the choice $\phi = (z-1)/4$ returns a spin-ice-like effective energetics, given by $E_s \sim q_s^2$, that explains the ice rule of colloidal ice in simple lattices. When the lattice has multiple coordinations, however, there is no single value of $\phi$ that can generate an effective ice-like energetics for vertices of more than one coordination. For $z = 4,3$, charge conservation imposes $\phi = (3-1)/2 = 1/2$ and thus the effective energetics maintains the ice rule on $z = 3$ vertices. On $z = 4$ vertices, however, it ascribes the same effective energy to the negative ($q = -2$) monopoles and to the ice rule ($q = 0$) vertices[47]. This explains why those are the only vertices seen in our simulations and experiments.

Another way to understand the same effect was reported recently[48]. It was demonstrated theoretically that a decimated particle-based ice is energetically equivalent to a spin ice stuffed with extra, negative topological charges, placed at the two ends of each decimated trap. For a spin system this implies accumulation of positive charge on decimated vertices (in the current system, $z = 3$ vertices) and, because the total charge of the occupied traps must be zero, the consequent formation of negative monopoles on undecimated vertices (here $z = 4$) vertices.

From these considerations one can quantitatively predict the charge transfer and thus the vertex statistics using a very simple entropic argument. In Methods we show how to obtain a very simple entropy density based on the simplifying assumption of uncorrelated vertices. Such entropy depends only on the amount of the charge transferred $q_{z_4}$. By maximizing that entropy, we obtain $q_{z_4}$, which is plotted in Fig. 5(c). Since the vertex populations are controlled by the charge transfer, we also obtain all the other relevant nonzero quantities.

Our purely entropic predictions in Fig. 5, obtained without any fitting parameters, agree remarkably well with the numerical results of simulations of large lattices with periodic boundary conditions. Small deviations from the theoretical predictions at low decimation come from the simplifying hypothesis of uncorrelated vertices. While such an assumption works well in a disordered ensemble, it is expected to produce deviations from the numerical results at low decimation $\eta < 1$ where the system is still largely in an ordered state (see also Fig. 4).

We also find very good agreement with the experimental data. There, deviations occur due to the limited size of the system which inevitably causes some charges to be confined at the boundaries. The agreement confirms the purely entropic nature of the ice rule fragility in this system as explained in the previous subsection.

The focus of this work is to prove the ice rule fragility in particle-based ices. We now describe another interesting effect brought about by the topological charge entropy which invites further study: the breakdown of order in the system. Undecimated square ice is antiferromagnetically ordered[11,23]. It has been recently proved theoretically[48] that under decimation, square ice crosses through a structural transition to disorder. At zero temperature, the system is ordered below a critical decimation, and disordered above it. The exact value of the critical decimation has not been computed exactly (though our current numerical analysis places it at about 10%, or $\xi$ ~0.1); it corresponds to a percolation threshold in the dimer model on which our decimation protocol is based[48], a subject currently under numerical study by others[50]. The formalism reported above and in[47], relying on uncorrelated charges, applies to disordered manifolds, and therefore does not predict such a transition.

Clearly in the low-decimation, ordered phase there no topological charge transfer and no ice-rule fragility is predicted. Instead we see in Fig. 4 that the breakdown of the ice rule is continuous as a function of decimation. The reason is that our numerical model is a Brownian dynamics simulation devised to faithfully reproduce the experimental apparatus; like the latter, it does not reach the true ground state but enters a state close to it[17,23]. For instance, the first panel of Fig. 4 (zero decimation) shows the presence of ordered domains separated by domain walls. These contain topological charges, though their net charge is zero. It is easily proven that the negative and positive charges of these excitations must alternate along such domain walls in the absence of decimation, giving no net charge transfer. At low decimation, as shown in Fig. 4, the domain walls pin to the decimated plaquettes, which preferentially carry negative charge (a situation analogous to that of doped colloidal ice[19]). The charge alternation is thus broken on those pinning sites, generating a net topological charge on the $z = 4$ sector.

This mechanism is better understood by considering how the disorder of the ground state sets in above critical decimation. Following ref.[48], the residual entropy of the ground state and the topological charge transfer are associated with the appearance of emergent lines composed solely of negative charges, connecting $q = -2$ monopoles on the $z = 4$ vertices that belong to decimated plaquettes. These lines must thread through nearest neighboring decimated plaquettes, and thus they exist only at decimations large enough that the decimated plaquettes percolate. Below that threshold, in the ground state, no such emergent lines exist, no topological charge transfer occurs, and the system remains ordered. The lines can still appear as small energy excitations, where they must include not only $q = -2$ monopoles, but also ice rule vertices that do not belong to the ground state (more precisely the fourth vertex from the left in the second line of Fig. 2(b), depicted as a gray disk). These excited emergent lines can still thread through decimated plaquettes even when the latter are not percolating. As the decimation is further reduced, such lines become simple domain walls, shown pinned to the decimated plaquettes in Fig. 4 at low decimation ($\xi = 0.02, 0.06$).

Thus we have the following interesting picture: in the equilibrium ground state below a critical decimation, the system is ordered and obeys the ice rule, while above it the system is disordered and the ice-rule is violated[48]. In slightly excited states at low decimation, the system forms ordered domains separated by lines pinned to the decimated plaquettes. As the decimation increases, these domains shrink, until at the percolation threshold for the decimated plaquettes, no order is discernible. This mechanism is apparent in the panels of Fig. 4 and consistent with previous observations in doped colloidal ice[19]. While the domains of the system are strongly correlated at low decimation, the domain walls are not, explaining why our uncorrelated-charge treatment above can capture the numerical data even at low correlation, and testifying to the solidity of the use of topological charges as degrees of freedom for describing this phenomenon. Indeed it was suggested[48] that dynamical arrests of the topological charges, though not necessarily of the colloids, could occur. The associated weak ergodicity breaking might thus make it impractical to observe the predicted structural transition in real systems, an issue that invites further theoretical and experimental investigation.

## Discussion
We have added a chapter to the long history of the ice rule by demonstrating the fragile nature of particle-based ice. Previously, in magnetic spin ice, the ice rule had proven remarkably robust to the introduction of all types of structural defects, doping, or dislocations. In contrast, in colloidal ice, the ice manifold is of a collective, non-local origin and can be destabilized by topology, leading to the spontaneous formation and accumulation of extensive topological charges which can rearrange and screen.

Our work has implications beyond ice rule systems for understanding classical topological phases. For example, the ensemble of ice-rule-obeying configurations, most of which are disordered, is called a Coulomb phase and is an example of a topological phase[51–53]. Topological states are increasingly studied in classical settings and in soft matter systems[31,54–56], where they are generally associated with stability produced by topological protection. In this context, our work poses a set of questions regarding whether such topological protections are robust to dilution or are instead fragile.

Furthermore, from an applied perspective, the possibility of controlling the dynamics and flow properties of topological charges via lattice decimation can inspire the engineering of novel dissipation-free magnetic storage and logic devices at the micro and nano-scale. For example, in domain wall engineering, interfaces of mixed coordination would be charged and possibly semi-permeable to defects, while in driven kinetics, entropically spontaneous charges could be suppressed or enhanced by an ac driving field.

More fundamentally, geometric frustration is a topic of considerable interest, as it encompasses a large class of physical systems in condensed matter and beyond, including biological systems. The ice-rule[1,4] has played a fundamental role in frustration, inspiring celebrated theoretical models[57,58] and appearing in an increasing number of physical and non-physical systems[38,59,60]. Our findings open a path toward a different phenomenology in geometrically frustrated, ice-rule based systems that is completely absent in traditional spin ices.

## Methods

**Experimental system**. The samples used in this work were prepared following a process similar to that described in Refs.[23,24]. In brief (see Fig. 6), we use soft lithography to create two-dimensional square lattices of bistable topographic traps, each 21 μm in length and 7 μm in width. The lattice constant is 29 μm. Each double well has a lateral confinement of depth ~3 μm and contains a central hill with average elevation $\langle h \rangle = 0.32 \pm 0.08$ μm (see Fig. 6). Within the trap we deposit paramagnetic colloidal particles that are 10 μm in size and that have a magnetic volume susceptibility of $\chi_m = 0.023 \pm 0.002$. The particles were diluted in highly deionized water and allowed to sediment above the sample due to density mismatch. To load one particle per double well, we use optical tweezers made with a $\lambda = 975$ nm, $P = 330$ mW butterfly laser diode focused by an oil immersion Nikon Plan Fluor 100× objective ($NA = 1.4$). The optical tweezers is mounted in a custom inverted optical microscope equipped with a white light illumination LED (MCWHL5 from Thorlabs) and a CCD camera (Basler A311f). The external magnetic field is applied with a custom-made coil oriented perpendicular to the sample cell and connected to a computer controlled power amplifier (KEPCO BOP-20 10M).

**Soft lithographic structures**. For the lithographic fabrication procedure, we write a square lattice of double wells on a mask made by a 5-inch glass wafer and covered with a 500 nm layer of Cr. Direct Write Laser Lithography (DWL 66, Heidelberg Instruments Mikrotechnik GmbH) was used for this purpose, based on a 405 nm laser diode and working at a speed of 5.7 mm²min⁻¹. The structures are designed using commercial software (CleWin 4, PhoeniX Software). Each double well is drawn on the mask as a stadium-shaped transparent region, with a small rectangular opaque spot in the center. The outer region has a length of 21 μm and a width of 7 μm, while the spot covers an area of 3 μm × 2 μm. The Cr mask is then used to etch the microfeatures on a 2.8 μm layer of photoresist AZ-1512HS (Microchem, Newton, MA). The photoresist is deposited on top of a 100 μm thick glass coverslip by spin coating (Spinner Ws-650Sz, Laurell) at 500 rpm for 5 s and afterwards at 1000 rpm for 30 s, both steps with an acceleration of 500 rpm/s. Different thicknesses of the photoresist could be obtained by varying the rotating speed; however, we find that ~3 μm works well to create topographical traps capable of capturing the particles within the double wells for most of the applied fields. After the deposition process, the photoresist is irradiated with UV light passing through the Cr mask for 6 s at a power of 25 mW/cm² (UV-NIL, SUSS Microtech). The light passing through the motifs of the mask uncrosslink only the exposed part of the photoresist. The exposed parts are then eliminated by submerging the film in a AZ726MF developer solution (Microchem, Newton MA). At this thickness, the size of the spot is too small for the lithographic process, and results in a small hill with a lower height at the center of the islands.

**Numerical simulation**. We conduct Brownian dynamics simulations of the decimated colloidal ice system comprised of magnetically interacting colloids with a radius of 5.15 μm placed in an array of $N_t = 50 \times 50 \times 2 = 5000$ etched double-well grooves arranged in a square lattice with a lattice constant of 29 μm giving a total of $N_v = 2500$ vertices. We use periodic boundary conditions in both the $x$ and $y$ directions. The double-well trap consists of two halves of a parabolic well joined by an elongated part. The particle in either parabolic half is tethered to the center of the well with spring constant of 1.212 pN/μm. Along the elongated part of the pinning site, this same tethering force confines the particle perpendicularly, while a repulsive force with a spring constant of $k_m = 0.352$ pN/μm repels the particle from the middle of the pinning site, reaching a maximum value of $F_m = 1.758$ pN in the middle of the pin and vanishing as it reaches the center of either well. These combined substrate forces acting on particle $i$ are written as $F_s^i$. Magnetization of the particles in the $z$ direction produces a repulsive particle-particle interaction force $F_{pp}(r) = A_c/r^4$ with $A_c = 3 \times 10^6 \chi_m^2 V^2 B^2/(2\pi\mu)$ for particles a distance $r$ apart. Here $\chi_m$ is the magnetic susceptibility, $V$ is the particle volume, $B$ is the magnetic field in mT, and all distances are measured in μm. This gives $F_{pp} = 7.231$pN for $r = 20$ μm at $B = 50$ mT, the maximum field we consider. The dynamics of colloid $i$ are obtained using the following discretized overdamped equation of motion:

$$\frac{1}{\mu}\frac{\Delta r_i}{\Delta t} = \sqrt{\frac{2}{D\Delta t}}k_B T N[0,1] + F_{pp}^i + F_s^i \qquad (1)$$

where the diffusion constant $D = 7000$ nm²/s, the mobility $\mu = 1.729$ μm/s/pN and the simulation time step $\Delta t = 1$ms. The first term on the right is a thermal force consisting of Langevin kicks of magnitude $F_T = 2.163$ pN corresponding to a temperature of $t = 20\,°C$ (when $N[0,1] = 1$). Here, $N[0,1]$ denotes a random number drawn from a normal (Gaussian) distribution with a mean of 0 and a standard deviation of 1. Each trap is initially filled with a single particle placed in a randomly chosen well, or left empty in the case of decimation. We increase $B$ linearly from $B = 0$ mT to $B = 50$ mT, consistent with the experimental range. We average the results over 10 simulations performed with different random seeds.

**Computation of theoretical curves**. We let $N_{z_4,s}$, $N_{z_3,s}$ denote the number of vertices containing $s$ colloids and having coordination $z = 4, 3$, respectively. From this, $n_{z_4,s} = N_{z_4,s}/N_{z_4}$ and $n_{z_3,s} = N_{z_3,s}/N_{z_3}$ are the relative frequencies of vertices with $s$ colloids in each sector. Finally, we write $q_{z_4}, q_{z_3}$ for the total densities of topological charge on the sublattices $z = 4$, $z = 3$, which are given by $q_{z_4} = \sum_{s=0}^{4} q_s n_{z_4,s}$, $q_{z_3} = \sum_{s=0}^{3} q_s n_{z_3,s}$.

As explained above and demonstrated in Ref.[47], at equilibrium all the $z = 4$ vertices are expected to be either obeying the ice rule—and thus of type 2-in ($q = 0$)—or to break the ice rule as lowest charge monopoles—and thus of type 1-in (q = −2). In contrast, all the $z = 3$ vertices must obey the ice rule, meaning that they are of type 2-in ($q = 1$) and 1-in ($q = -1$). The $z = 3$ vertices screen the extra charge by changing the relative admixture of ±1 charges. From the conservation of topological charge, we obtain the constraint

$$n_{z_4,1} = \eta\left(n_{z_3,2} - 1/2\right), \qquad (2)$$

which implies that a complete transfer of topological charge $\left(n_{z_4,1} = 1, n_{z_4,2} = 0\right)$ is possible in principle when $\eta \geq 2$.

Since configurations corresponding to partial charge transfer are in general entropically favored, the charge transfer is mostly entropic. Consider a charge transfer between two vertices of different coordination as in Fig. 1-Methods. If we ignore the small energy differences between ice-rule vertices, on both vertices the energy depends on the number of colloids in the vertex, and the charge transfer does not change the energy. To demonstrate the entropic nature of the charge transfer, we have shown in the main text that the ensemble can be quantitatively predicted by a purely entropic argument, which we report here. Consider the entropy

$$s = n_{z_4,1}\ln\left(n_{z_4,1}/4\right) + n_{z_4,2}\ln\left(n_{z_4,2}/2\right) \\ + \eta[n_{z_3,1}\ln\left(n_{z_3,1}/3\right) + n_{z_3,2}\ln\left(n_{z_3,2}/3\right)], \qquad (3)$$

where the denominators within the logarithms corresponds to the multiplicity of the respective vertex configurations at the numerators. We can minimize the entropy (3) under the constraints $n_{z_4,1} + n_{z_4,2} = 1$, $n_{z_3,1} + n_{z_3,2} = 1$, and Eq. (2). We thus obtain the density of topological charge per unit of $z = 4$ vertex $q_{z_4} = -2n_{z_4,1}$ as

$$q_{z_4} = \frac{1}{2}\left(\sqrt{9\eta^2 + 8\eta + 16} - 3\eta - 4\right), \qquad (4)$$

plotted in Fig. 4(c) of the main text, and which is smaller in absolute value than the maximal charge $q_{tot}^{max}$ permitted by the geometry, given by $q_{tot}^{max} = -\eta$ if $\eta \leq 2$ and $q_{tot}^{max} = -2$ if $\eta \geq 2$. Indeed, Eq. (4) gives $q_{z_4} \to -4/3$ as $\eta \to \infty$. Knowledge of charge transfer from Eq. (4) allows us to obtain all other relevant nonzero quantities: $q_{z_3} = -q_{z_4}/\eta$, $n_{z_4,1} = -q_{z_4}/2$, $n_{z_4,2} = 1 - n_{z_4,1}$, which are plotted in Fig. 4 of the main text.

## Data availability

All relevant data are available from the authors.

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

## Acknowledgements

D.Y.L, A.O.A. and P.T. acknowledge support from the ERC Starting grant "DynaMO" (No. 335040). A.O.A. acknowledges support from the "Juan de la Cierva" program (FJCI-2015-25787). P.T. acknowledge support from the Spanish MINECO (FIS2016-78507-C2) and DURSI (2017SGR1061). The work of C.R., C.J.O.R., and C.N. was carried out under the auspices of the National Nuclear Security Administration of the U.S. Department of Energy at Los Alamos National Laboratory under Contract No. DEAC52-06NA25396. C.R. and C. N. wish to thank LDRD at LANL for financial support through grant 20170147ER.

## Author contributions

C.N. originated the idea, provided its theoretical understanding, and supervised the project. A.L. performed the early numerical exploration to assess viability of an experiment, followed by full-scale, comprehensive simulations on large systems, in collaboration with C.R. and C.J.O.R. A.O.A. and D.Y.L. performed the experiment with equal contribution, under the guidance of and in collaboration with PT. All coauthors contributed equally to the analysis of numerical and experimental outcomes. C.N. drafted the

manuscript with help from C.J.O.R. All coauthors contributed equally to the finalization of the manuscript, figures and supplementary information.

## Additional information

**Competing interests:** The authors declare no competing interests.

