## [Peer Review File · Nature Communications]

Reviewers' comments:

Reviewer #3 (Remarks to the Author):

I have read the revised manuscript, the reports of both reviewers on the original manuscript, and the response of the authors to these reports. I also glanced at Ref 32 (Nisoli, New J Phys 2014), which is a key reference for judging the novelty of the present paper.

In terms of content, my judgment is that the manuscript describes work that is novel, substantial and important enough to merit publication in Nature Communications. The theory presented here has clearly developed compared to what is included in Ref 32, and the present paper provides a detailed quantitative investigation of the phenomena at hand, including comparisons with numerical simulations and with experiments, which were absent in Ref 32.

My main concern is the presentation. I feel that even after this resubmission following the comments of Reviewer #1, the present presentation is still too technical. Condensed matter physicists that are not specifically familiar with recent work on such spin ices, could find the paper difficult to follow, and I urge the authors to work on better presenting in simpler terms what are they about to do before getting too technical. Also, readers from slightly more distant fields would probably not even manage to understand the main message of the paper.

A few additional specific comments:

The theory and simulations assume only nearest neighbor interactions. What is the justification for neglecting interactions with further neighbors? Specifically for colloidal systems such as those studied here, the physical interactions between particles is long ranged. Not only that, but due to the particle nature of this system (which is the main point of the paper), distances to neighbors that are logically or topologically higher order neighbors might in fact be not that much larger than distances to nearest neighbors. This is due to the fact that each particle in each double well may move between its two distinct positions, which have very different distances to the particle positions in nearest and next nearest neighbor traps.

The main point of the paper seems rather simple (yet probably overlooked in the past and thus constitute a substantial advancement to the field, but also to condensed matter in general). Yet to demonstrate this point, the authors go via a rather complex path, both in terms of the systems studied and in terms of the analysis employed. Isn't there a simpler way to identify and demonstrate the essence of the effects that the authors are suggesting?

The main point of the paper seems appealing to condensed matter physics in general, especially given the interest in topological phenomena in electronic condensed matter systems. Clearly, the phenomena studied here are different and specific to the spin ice systems at hand. Nonetheless, the paper could be made stronger, if more general results were obtained that may be linked to other current research activities on topology in condensed matter.

Finally, the paper requires much editing to fix language and typos.

Reviewer #3 (Remarks to the Author):

I have read the revised manuscript, the reports of both reviewers on the original manuscript, and the response of the authors to these reports. I also glanced at Ref 32 (Nisoli, New J Phys 2014),

which is a key reference for judging the novelty of the present paper.

In terms of content, my judgment is that the manuscript describes work that is novel, substantial and important enough to merit publication in Nature Communications. The theory presented here has clearly developed compared to what is included in Ref 32, and the present paper provides a detailed quantitative investigation of the phenomena at hand, including comparisons with numerical simulations and with experiments, which were absent in Ref 32.

We thank the referee for the effort in evaluating the manuscript and for considering it novel, substantial, and deserving publication in Nature Communications.

My main concern is the presentation. I feel that even after this resubmission following the comments of Reviewer #1, the present presentation is still too technical. Condensed matter physicists that are not specifically familiar with recent work on such spin ices, could find the paper difficult to follow, and I urge the authors to work on better presenting in simpler terms what are they about to do before getting too technical. Also, readers from slightly more distant fields would probably not even manage to understand the main message of the paper.

We agree with the referee that the exposition was unnecessarily technical, which might limit its impact. We have completely rewritten the introduction, and we feel that we have better situated our results in the much broader context of constrained disorder in frustrated systems and the role played by the ice rule. We have also summarized the main results in the introduction, guiding the reader through the figures of the paper, and explained them in more intuitive and heuristic terms.

We have confined the more technical matters to the Results section, but there we have also expanded definitions and explanations to make the exposition more accessible. We thank the referee for helping us improving the manuscript.

A few additional specific comments:

The theory and simulations assume only nearest neighbor interactions. What is the justification for neglecting interactions with further neighbors? Specifically for colloidal systems such as those studied here, the physical interactions between particles is long ranged. Not only that, but due to the particle nature of this system (which is the main point of the paper), distances to neighbors that are logically or topologically higher order neighbors might in fact be not that much larger than distances to nearest neighbors. This is due to the fact that each particle in each double well may move between its two distinct positions, which have very different distances to the particle positions in nearest and next nearest neighbor traps.

That is an excellent observation and indeed the long range ($1/r^3$) of the interaction explains the screening of charges. Though it has not been proven in any conclusive mathematical way, it seems established at least heuristically that in these systems (both particle-based and spin-based) the nearest neighbor interaction controls the onset of the ice rule whereas the the next nearest neighbor interactions can produce further ordering within a disordered ice manifold. In magnetic systems this has been demonstrated e.g. by Muller and Moessner *Physical Review B* 80.14 (2009): 140409. Some of the authors have recently shown, both numerically (Libál, A., et al. *Physical Review Letters* 120.2 (2018): 027204.) and analytically (C. Nisoli *Physical Review Letters* 120.16 (2018): 167205.) that this is also the case in particle-based ices, assuming that the interaction is long ranged. However, whether the expansion of the energy in terms of

successive neighbor contribution translates in general into nested phases of progressively lower entropy is indeed not demonstrated. We have elaborated on such issues in the manuscript where we deal with the charge screening.

The main point of the paper seems rather simple (yet probably overlooked in the past and thus constitute a substantial advancement to the field, but also to condensed matter in general). Yet to demonstrate this point, the authors go via a rather complex path, both in terms of the systems studied and in terms of the analysis employed. Isn't there a simpler way to identify and demonstrate the essence of the effects that the authors are suggesting?

The most succinct explanation is the following. The vertex energetics of particle-based ices, unlike those of magnetic spin ices, do not possess Z_2 symmetry. However, their collective energetics does possess such symmetry (leading to the ice rule) but only for trivial lattices endowed with certain symmetries. Decimation breaks that symmetry. That can be demonstrated rigorously and indeed one of us did so recently (C. Nisoli Physical Review Letters 120.16 (2018): 167205.). We have mentioned and added the reference.

The main point of the paper seems appealing to condensed matter physics in general, especially given the interest in topological phenomena in electronic condensed matter systems. Clearly, the phenomena studied here are different and specific to the spin ice systems at hand. Nonetheless, the paper could be made stronger, if more general results were obtained that may be linked to other current research activities on topology in condensed matter.

We agree with the referee that our results have a generality that should be stressed in the context of classical topological states.

Indeed, topological features are generally associated with a novel kind of stability. We can see, for instance, that ice-manifolds can be, depending on the geometry, Coulomb phases which are in general considered topological states, whose disorder is labeled by an emergent gauge field, and whose excitations are topological charges. Here we show indeed that a topological state can be destabilized, and in this case it is destabilized precisely by the generation of those topological charges. We have added a discussion on that matter, as well as proper references.

Finally, the paper requires much editing to fix language and typos.

We have fixed that. We thank the referee for the considerable help in improving the manuscript.

REVIEWERS' COMMENTS:

Reviewer #3 (Remarks to the Author):

The revised manuscript is substantially improved, however before acceptance there are a few points which the authors should address:

Why is q termed a topological charge? What topological property does it relate to?

In the caption to figure 2, the authors say that vertices with the same number of in colloids have the same energy. This does not seem to be precise and is at odds with what is written subsequently in the same caption, that of the 2-in configurations, one has a lower energy than the others.

The counting arguments presented at the bottom of the left column page 3 ignore boundary effects. Properly accounting for the finite size of the system should modify the expressions obtained there.

Page 3, right column - When saying that vertices with $z=3$ have odd charges, zero should be removed from the list of odd numbers.

I couldn't find any reference in the main text or in the methods section about the system size in the experiments.

Page 4, right column - It would be good to briefly explain in words what ingredients or assumptions enter the theoretical prediction which is subsequently described only in the methods section.

Page 5, right column, and following paragraph in page 6 - It is not clear how are the measurements reported here related to dynamics.

Figure 5 displays a reasonable agreement between experiments, simulations and theory. Yet there are some differences between the three, and a discussion of the possible origins of these differences would be in place.

Methods - Numerical simulations - What are $N[0,1]$ and F_s^i in Eq. (1)? Where is F_T that is referred to in the text used?

The following typographical errors should be corrected:

Caption to figure 3 - theoloidal

Page 7, left column - thets

Page 7, left column - is a n example

References 27, 28, 33 - O'Brien

Reviewer #3 (Remarks to the Author):

The revised manuscript is substantially improved, however before acceptance there are a few points which the authors should address:

Why is q termed a topological charge? What topological property does it relate to?

The charge is topological insofar as it depends upon the connectivity of the system, and its definition does not change for continuous deformations of the lattice. It is therefore a topological invariant for the vertex configuration. As such it generally provides the best language to describe these materials because the ice manifold corresponds to local minimization of the topological charge. Of course, interactions in these materials are geometrical and therefore not topologically invariant: hexagonal and brickwork spin ice are topologically equivalent yet their spin ensemble differ inasmuch as the former is disordered and the latter ordered. However, in both cases the low energy state corresponds to minimal local topological charge. We have added an explanation of this point to the text.

In the caption to figure 2, the authors say that vertices with the same number of in colloids have the same energy. This does not seem to be precise and is at odds with what is written subsequently in the same caption, that of the 2-in configurations, one has a lower energy than the others.

The referee is correct, and we have removed that statement. That was an oversight and we are grateful to the referee for catching it.

The counting arguments presented at the bottom of the left column page 3 ignore boundary effects. Properly accounting for the finite size of the system should modify the expressions obtained there.

The referee is correct, but in fact we always consider the thermodynamic limit in our theoretical expressions. Indeed, we explain later on (but also see ref [45]) that the ice rule in particle-based ice is only recovered in the thermodynamic limit in which boundaries are negligible. For finite systems, positive topological charge is pushed to the boundaries, creating violations of the ice rule in the bulk. The density of such bulk defects scales at least as fast as the reciprocal length of the boundaries. We have added a clarifying statement addressing this point on page 3.

Page 3, right column - When saying that vertices with $z=3$ have odd charges, zero should be removed from the list of odd numbers.

We are very grateful to the referee for catching this typo, which we have corrected.

I couldn't find any reference in the main text or in the methods section about the system size in the experiments.

We now state in the text on pages 3-4 that the experimental system consists of 11×8 vertices corresponding to a total of $N_t=195$ traps in the undecimated case.

Page 4, right column - It would be good to briefly explain in words what ingredients or assumptions enter the theoretical prediction which is subsequently described only in the methods section.

We have consolidated the theoretical treatment in a subsection of results, in which we explain in more simple terms the basic ingredients. We have also added considerably to the discussion of Figure 4 and to how transfer of topological charges lead to a disordered manifold, and related it to more recent literature.

Page 5, right column, and following paragraph in page 6 - It is not clear how are the measurements reported here related to dynamics.

Indeed, the measures do not relate to dynamics. While the simulations are Brownian dynamics, here we only report the final convergence state. We have removed the word dynamics.

Figure 5 displays a reasonable agreement between experiments, simulations and theory. Yet there are some differences between the three, and a discussion of the possible origins of these differences would be in place.

There is some disagreement at very low decimation between theory and numerical results. This is due to the simplicity of the theory which considers the vertices as completely uncorrelated. This is reasonable at higher decimation: indeed, in another work [45] we demonstrate how decimation brings in disorder, but that this can break down at low decimation. We indeed find that disorder kicks in at relatively low decimation (less than $\eta=0.9$, from Fig. 4), precisely in the regime where the small deviation appears in Fig. 5. In particular, at such low decimation, numerics and experiment show the appearance of charges $+3$ on $z=3$ vertices. This is a correlation effect: there are not enough $z=3$ vertices to adsorb the available charge and they are typically all surrounded by $z=4$ vertices, and as a result each $z=3$ vertex can individually adsorb more charge. This mechanism is not considered in the theoretical treatment. Though it could be easily introduced, we decided not to cumber the formalism to map more finely a parameter region which is not the most interesting.

As we state in the paper, deviations of the experimental data are more significant. The origin of such deviations is twofold. Firstly, the sample is of finite size, so there is a formation of negative -2 charges on the $z=4$ vertices. Those charges also appear because the system is not completely equilibrated, as was the case in previous experiments on the same system (cited in the text).

We have extended the discussion of these points in the text.

Methods - Numerical simulations - What are $N[0,1]$ and F_s^i in Eq. (1)? Where is F_T that is referred to in the text used?

We now provide a definition in the numerical simulation portion of the methods section, and also indicate that F_s^i is the combined substrate forces acting on particle i . We also clarify that F_T is simply the value of the prefactor of the $N[0,1]$ term.

The following typographical errors should be corrected:

Caption to figure 3 - theoloidal

Page 7, left column - thets

Page 7, left column - is a n example

References 27, 28, 33 - O?Brien

We have corrected these typos and we wish to thank the reviewer for the careful reading and for helping us improve the manuscript.